# Control of neural crest multipotency by Wnt signaling and the Lin28/*let-7* axis

**Debadrita Bhattacharya, Megan Rothstein, Ana Paula Azambuja, Marcos Simoes-Costa\***

Department of Molecular Biology and Genetics, Cornell University, Ithaca, United States

**Abstract** A crucial step in cell differentiation is the silencing of developmental programs underlying multipotency. While much is known about how lineage-specific genes are activated to generate distinct cell types, the mechanisms driving suppression of stemness are far less understood. To address this, we examined the regulation of the transcriptional network that maintains progenitor identity in avian neural crest cells. Our results show that a regulatory circuit formed by Wnt, *Lin28a* and *let-7* miRNAs controls the deployment and the subsequent silencing of the multipotency program in a position-dependent manner. Transition from multipotency to differentiation is determined by the topological relationship between the migratory cells and the dorsal neural tube, which acts as a Wnt-producing stem cell niche. Our findings highlight a mechanism that rapidly silences complex regulatory programs, and elucidate how transcriptional networks respond to positional information during cell differentiation.

DOI: https://doi.org/10.7554/eLife.40556.001

## Introduction

The process of cell differentiation is characterized by major shifts in the molecular programs that control cell identity. This requires the coordination of activating and repressive regulatory mechanisms, which act together to drive overarching changes in gene expression (*Davidson, 2009*). In the past decades, substantial progress has been made in understanding how the activation of lineage-specific genes generates distinct cell types. In contrast, we know far less about how progenitor cell identity is silenced during cell fate commitment. Inhibitory mechanisms are crucial for terminal differentiation since silencing of multipotency networks precedes activation of lineage-specific factors and chromatin remodeling (*Kalkan and Smith, 2014*; *Moris et al., 2016*). Repression also plays an essential role in adult tissue homeostasis. Anomalous reactivation of embryonic regulatory programs in somatic tissue has been shown to underlie tumorigenesis and metastasis (*Nieto, 2013*; *Kaufman et al., 2016*). Despite their importance in development and disease, the mechanisms that suppress stemness and multipotency remain elusive.

Here, we used the cranial neural crest as a model to examine how stem cell identity is regulated during differentiation. This multipotent cell population contributes to numerous tissues and organs in vertebrate embryos, including the craniofacial skeleton, peripheral nervous system, and pigmentation of the skin (*Le Douarin, 1982*). Neural crest cells delaminate from the neural tube to engage in extensive migration throughout the embryo (*Figure 1a–b*). As these cells move away from the neural tube, they undergo drastic changes in their transcriptional identity, and transition from multipotent progenitors to committed cell types. The formation and differentiation of the neural crest is controlled by a complex gene regulatory network, composed by multiple signaling pathways, transcription factors, and epigenetic modifiers (*Simões-Costa and Bronner, 2015*; *Meulemans and Bronner-Fraser, 2004*; *Betancur et al., 2010*). Expression of the early components of this network defines the neural crest stem cell population that resides within the dorsal neural tube (*Lignell et al., 2017*).

**\*For correspondence:**
simoescosta@cornell.edu

**Competing interests:** The authors declare that no competing interests exist.

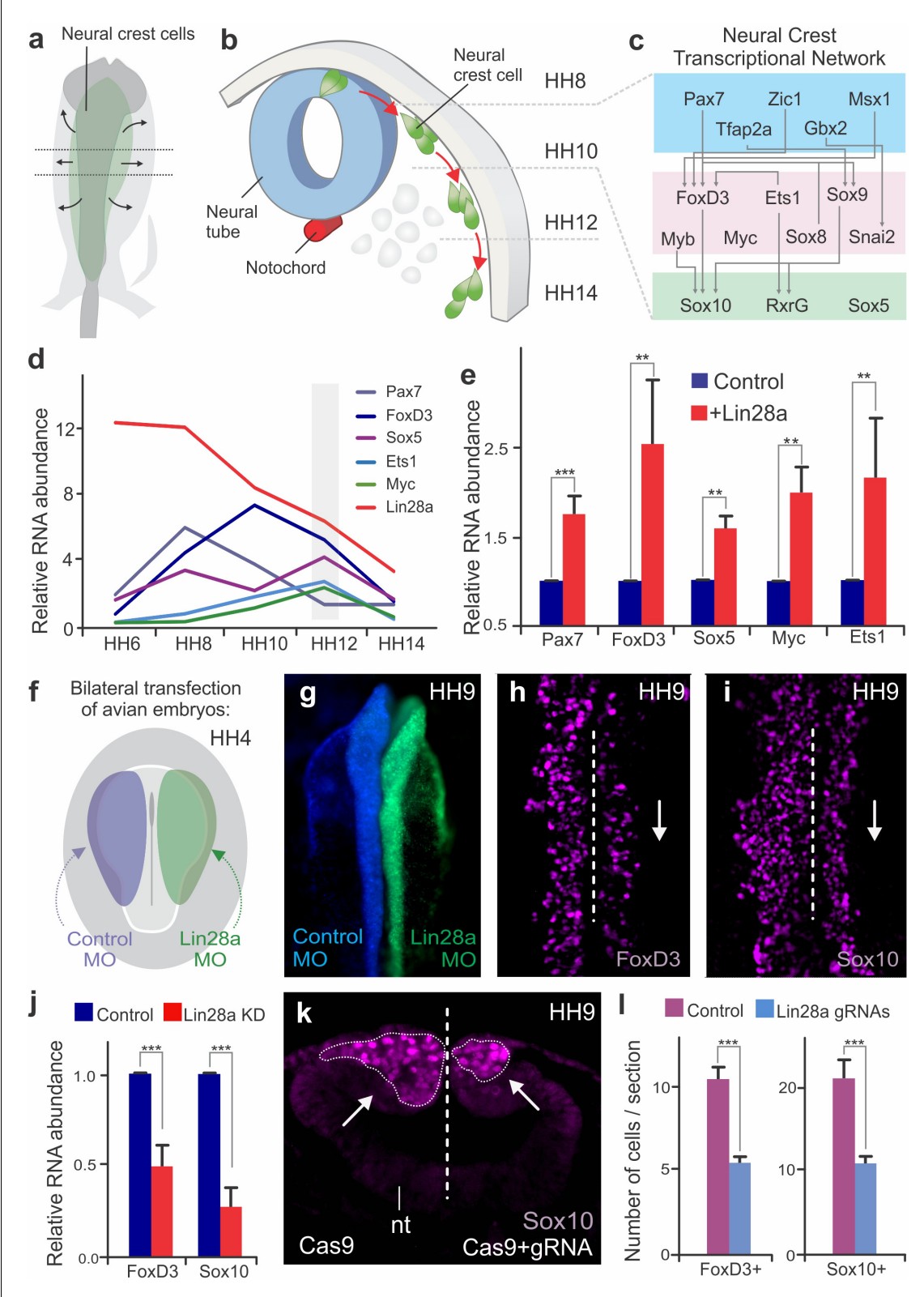

**Figure 1.** Changes in *Lin28a* levels impact neural crest development in vivo. (a–b) Neural crest migration during avian development. (a) Neural crest progenitor cells (green) are specified on dorsal folds of the neural tube (grey) during early development. (b) Transverse section of the neural tube showing the position of neural crest cells through development, as they progressively move away from the neural tube to differentiate. HH8 and HH14 are the earliest and latest developmental stages shown in the diagram, respectively. (c) A schematic of the early gene regulatory network composed of

*Figure 1 continued on next page*

*Figure 1 continued*

transcription factors involved in neural crest cells formation. (**d**) Expression levels of *Lin28a* and transcription factors of the early gene regulatory circuit, in sorted neural crest cells obtained from different stages. (**e**) Constitutive expression of *Lin28a* results in maintenance of multipotency genes in late neural crest cells. RT-PCR for *Pax7, FoxD3, Sox5, Myc and Ets1* comparing the expression of these genes in control *vs.* Lin28a overexpressing migratory neural crest cells. (**f**) Electroporation scheme for loss-of-function assays in which control reagent (blue) and targeted reagent (green) were injected in different sides of a HH4 chick embryo. (**g**) Dorsal whole mount view of HH9 embryo with Control MO on the left and Lin28a MO on the right. Immunohistochemistry for neural crest markers FoxD3 (**h**) and Sox10 (**i**) on Lin28a knockdown. Dotted line represents embryo midline (**j**) RT-PCR for *FoxD3* and *Sox10* transcripts in control vs Lin28a MO treated neural folds. (**k–l**) CRISPR-Cas9 mediated knockdown of Lin28a recapitulates the MO phenotype. (**k**) Transverse section showing Sox10 positive cells in control and knockdown sides of the embryo head, showing reduction in the number of neural crest cells (arrow). (**l**) Quantification of FoxD3+ and Sox10+ cells following CRISPR-Cas9 mediated knockdown of Lin28a. Error bars in (**e**), (**j**) and (**l**) represent standard error. HH: Hamburger and Hamilton developmental stages, MO: Morpholino.

DOI: https://doi.org/10.7554/eLife.40556.002

The following source data and figure supplements are available for figure 1:

**Source data 1.** The RT-PCR results of *Lin28a* over-expression and morpholino-mediated knockdown experiments and the quantitation of FoxD3 +and Sox10 +cells following CRISPR.

DOI: https://doi.org/10.7554/eLife.40556.007

**Figure supplement 1.** Expression patterns of *Lin28a* and *Lin28b* mRNA and Lin28a protein during early chick development.

DOI: https://doi.org/10.7554/eLife.40556.003

**Figure supplement 2.** Ectopic expression of *Lin28a* prevents silencing of early neural crest genes and delays differentiation.

DOI: https://doi.org/10.7554/eLife.40556.004

**Figure supplement 3.** Supporting data for *Lin28a* and *Lin28b* loss-of-function analysis.

DOI: https://doi.org/10.7554/eLife.40556.005

**Figure supplement 4.** Effects of Lin28a loss-of-function on cell death, proliferation and the morphology of cranial ganglia.

DOI: https://doi.org/10.7554/eLife.40556.006

This set of genes, which includes neural crest markers and pluripotency factors, endows the neural crest stem cells with their unique features, such as multipotency and self-renewal.

In this study, we have uncovered a regulatory circuit formed by Wnt signaling and the Lin28a/*let-7* axis that modulates stem cell identity of avian cranial neural crest cells. This circuit functions in a position-dependent manner, promoting multipotency in the early neural crest cells and controlling the transition to differentiation in the late-migrating cells. In the premigratory neural crest, canonical Wnt signaling directly activates *Lin28a* transcription, resulting in the inhibition of *let-7* miRNA activity. As neural crest cells migrate away from the Wnt source, *let-7* levels increase, and these miRNAs target and silence multiple components of the neural crest gene regulatory network. As a result, the early network collapses and stem cell identity is suppressed. Thus, we propose that the changes in the topological relationship between the neural crest and the dorsal neural tube drive the transition from multipotent stem cell to differentiated cell type. Our model integrates signaling, transcription and post-transcriptional regulation to clarify how positional information ultimately affects gene network topology and stem cell identity.

## Results

### Dynamic expression of Lin28a during neural crest development

The developmental program controlling the formation of neural crest cells is encoded in a modular transcriptional gene regulatory network (*Simões-Costa and Bronner, 2015*; *Meulemans and Bronner-Fraser, 2004*; *Betancur et al., 2010*) (*Figure 1c*), in which the early modules mediate neural crest induction and specification, while the later modules regulate their ability to migrate and differentiate into diverse cell types (*Simões-Costa and Bronner, 2015*; *Meulemans and Bronner-Fraser, 2004*). A large part of the early gene regulatory network (*Figure 1c*) consists of factors that maintain neural crest cells in a multipotent state, such as *Pax7*, *FoxD3*, *Ets1*, *Myc*, and *Sox5* (*Buitrago-Delgado et al., 2015*; *Le Douarin and Dupin, 2016*). These factors define the neural crest stem cell pool and are co-expressed in pre-migratory and early migrating cells (*Lignell et al., 2017*). To examine how stem cell identity changes during cell commitment, we quantified the expression levels of these factors during different stages of embryonic development. We labeled cranial neural crest cells with an enhancer of the *Tfap2a* gene (here referred as *Tfap2aE1*) (*Attanasio et al., 2013*), which

drives specific expression of reporter genes in the avian neural crest lineage, starting at HH6 and persisting until late migratory stages (HH16). We generated a *Tfap2aE1*-eGFP construct to FACS-sort pure populations of neural crest cells from different embryonic stages for RT-PCR analysis. At the stages corresponding to progenitor specification, we observed an increase in the mRNA levels of genes that are part of the neural crest stem cell signature, like *Pax7*, *FoxD3*, *Ets1*, *Myc*, and *Sox5* (*Figure 1d*). While the expression of these factors peaked at different time points, we observed that they were simultaneously downregulated during the late stages of migration. Such striking changes in the neural crest transcriptional network raised the intriguing possibility that a regulatory mechanism exists to silence stem cell identity at the onset of differentiation.

Our transcriptomic analyses of cranial neural crest cells (*Simoes-Costa and Bronner, 2016*; *Simões-Costa et al., 2014*) had previously shown that pluripotency factor *Lin28a* is strongly enriched in this cell population. To test if *Lin28a* is involved in the transition from multipotency to differentiation, we first examined its expression levels during neural crest development. *In situ* hybridization and immunohistochemistry analysis revealed that *Lin28a* was robustly expressed in the neural folds, premigratory and early migrating neural crest of the chick embryo (*Figure 1—figure supplement 1a–p*). Isolation of neural crest with the *Tfap2aE1* and RT-PCR quantitation showed strong enrichment of *Lin28a* mRNA during specification (*Figure 1—figure supplement 1q*). This was not observed for the other Lin28 gene present in the chick genome, *Lin28b* (*Tsialikas and Romer-Seibert, 2015*). The paralog was expressed at much lower levels and was not enriched in neural crest cells (*Figure 1—figure supplement 1q–r*). Interestingly, there was a marked reduction in *Lin28a* mRNA levels at later stages, concomitant with silencing of the neural crest stem cell genes (*Figure 1d*). These results show that *Lin28a* is dynamically expressed during neural crest development, and that it is most abundant in premigratory cells that reside within the neural tube.

## Lin28a promotes maintenance of neural crest stem cell identity

To verify whether *Lin28a* is involved in the regulation of neural crest stem cell identity, we used transient expression vectors to manipulate its levels in developing chick embryos. First, we investigated how sustained expression of *Lin28a* affects the silencing of the early neural crest genes at later stages of migration. Neural crest cells transfected with a *Lin28a* expression vector were sorted through FACS, and their expression profile was compared to wild-type neural crest through RT-PCR. This analysis revealed that migratory cells constitutively expressing *Lin28a* maintained expression of neural crest stem cell factors even at stages when these genes would normally be downregulated (*Figure 1e*). We confirmed these results by performing immunohistochemistry for Pax7 and FoxD3 following *Lin28a* gain-of-function (*Figure 1—figure supplement 2a–h*). This maintenance of early neural crest factors resulted in a delay in differentiation, as evidenced by a decrease in the expression of ectomesenchymal, neuronal and glial markers in late migrating cells (*Figure 1—figure supplement 2i*). This is consistent with the possibility that *Lin28a* acts to prevent the silencing of the multipotency network that precedes cell differentiation.

In the reciprocal experiment, we tested the effects of premature downregulation of *Lin28a* in neural crest stem cells. To this end, we performed bilateral electroporations to transfect embryos with a *Lin28a* translation-blocking morpholino on the right side (green), and a control morpholino (blue) on the left side (*Figure 1f*). This treatment resulted in a strong knockdown of *Lin28a* protein in the morpholino-transfected side of the embryo (*Figure 1—figure supplement 3a–d*). Loss of *Lin28a* resulted in decreased expression of neural crest markers *FoxD3 and Sox10*, as shown by immunohistochemistry (*Figure 1g–i*) and *in situ* hybridization (*Figure 1—figure supplement 3e–h*). Immunohistochemistry for phospho-histone H3 and Caspase-3 confirmed that this phenotype was not due to changes in cell cycle exit or cell death (*Figure 1—figure supplement 4a–c,f–g*). Microdissection of dorsal neural folds of targeted embryos allowed for the quantification of the phenotype with RT-PCR, which revealed a significant decrease in the expression of the neural crest markers (*Figure 1j*, *Figure 1—source data 1*). To confirm this phenotype, we employed CRISPR/Cas9 genome editing and RNA interference (using Dicer-substrate siRNAs, DsiRNAs) (*Kim et al., 2005*) to disrupt *Lin28a* expression (*Figure 1k–l*, *Figure 1—figure supplement 3i–m*). Consistent with the morpholino experiments, targeting the first exon of *Lin28a* with a pair of gRNAs resulted in a significant reduction in the number of *FoxD3* and *Sox10* positive cells (*Figure 1k–l*, *Figure 1—source data 1*). Finally, DsiRNA-mediated knockdown of *Lin28a*, but not of *Lin28b*, resulted in a significant decrease in the expression of *FoxD3* and *Sox10* (*Figure 1—figure supplement 3n–o*).

To evaluate possible long-term consequences of *Lin28a* knockdown in neural crest derivatives, we employed electroporation of morpholinos combined with a cornish pasty culture system (*Nagai et al., 2011*), which allows for long-term incubation of electroporated embryos *ex ovo*. Bilaterally-injected morphant embryos were cultured until stage HH15 so that we could observe the formation of cranial ganglia. Immunohistochemistry with neuronal marker Tuj1 revealed that knockdown of *Lin28a* resulted in dispersed ganglia with abnormal condensation. In particular, we observed a marked reduction in the maxillomandibular lobe of the trigeminal ganglion (*Figure 1—figure supplement 4j–o*). This phenotype is consistent with previous studies that highlight the requirement of neural crest cells for timely ganglion condensation and accurate establishment of neuronal connections (*Stark et al., 1997*; *Shiau et al., 2008*; *Hamburger, 1961*). Taken together, these functional experiments indicate that high levels of Lin28a during early development are necessary for neural crest specification, while its subsequent downregulation is required for silencing of progenitor identity.

## Lin28a regulates neural crest multipotency

Lin28a is a *bona fide* pluripotency factor that has been shown to underlie stemness and to drive reprogramming of somatic cells (*Yu et al., 2007*; *Zhang et al., 2016*). Our observation that Lin28a manipulation impacts the neural crest transcriptional network suggests that this factor may regulate some of the stem cell properties that characterize this cell population. To directly test if Lin28a promotes neural crest multipotency, we examined how manipulating its expression affects the developmental potential of individual progenitors, using single-cell lineage analysis (*Sieber-Blum and Cohen, 1980*; *Trentin et al., 2004*). Neural crest cells from quail embryos transfected with *Lin28a* (or the empty PCI-H2b:RFP vector) were cultured in sparse conditions, such that single progenitor could form clonal colonies of differentiated cells (*Figure 2a*). After a ten-day incubation period in a standard culture medium, individual cells gave rise to colonies that were composed of multiple differentiated derivatives, which were identified based on cell morphology and molecular markers (*Figure 2b–c*). The somatic cell types observed included the typical neural crest derivatives, such as neurons (N), glia (G), melanocytes (M), chondroblasts (C) and smooth muscle cells (S). Quantification of cell types in colonies from the control and experimental groups revealed a six-fold increase in multipotent (that generated four or more cell types) neural crest cells after transfection with a Lin28a expression vector (*Figure 2d*). Furthermore, while mock-transfected neural crest cells were biased towards a glial fate (60% of the total progeny), the experimental group had a more uniform distribution of derivatives (*Figure 2e*, *Figure 2—source data 1*). Consistent with this, classification of neural crest progenitors according to their developmental potentials revealed a larger percentage of multipotent progenitors (like GCMS progenitors, which give rise to glia, chondroblasts, melanocytes and smooth muscles), when compared to the control group (*Figure 2f*, *Figure 2—source data 1*). The results from this single-cell clonal analysis indicate that the changes in the transcriptional network observed following manipulation in the Lin28a levels affect specific properties of neural crest cells, such as their ability to give rise to multiple cell types.

## The Lin28a/let-7 axis modulates neural crest differentiation

Lin28a has been shown to regulate genes post-transcriptionally both directly by binding to mRNAs, or indirectly by inhibiting maturation of the *let-7* family of microRNAs, which are potent post-transcriptional repressors (*Newman et al., 2008*). To investigate if Lin28a function in neural crest is *let-7* dependent, we employed a *let-7 sensor* to survey microRNA activity during different stages of development (*Figure 3a*). In this construct, a reporter gene with a destabilization domain (mCherry-PEST) was placed upstream of multiple *let-7* target sites, such that increased *let-7* activity results in decreased mCherry fluorescence. By transfecting embryos with the sensor, we observed that although neural crest stem cells have low *let-7* activity, it continuously increases as the cells migrate away from the neural tube (*Figure 3b*). Consistent with this, RT-PCR analysis revealed a significant increase in the expression levels of *let-7* miRNAs from the premigratory (HH8) to the late-migrating (HH12) neural crest (*Figure 3c*). Since *let-7* activity and expression was inversely correlated with *Lin28a* expression levels (*Figure 1d*), we tested the effect of Lin28a knockdown on *let-7* activity in the neural crest. The results show that in the absence of Lin28a, *let-7* activity increased, as evidenced by a reduction in *let-7* sensor expression (*Figure 3d–e*) and direct measurement of multiple mature

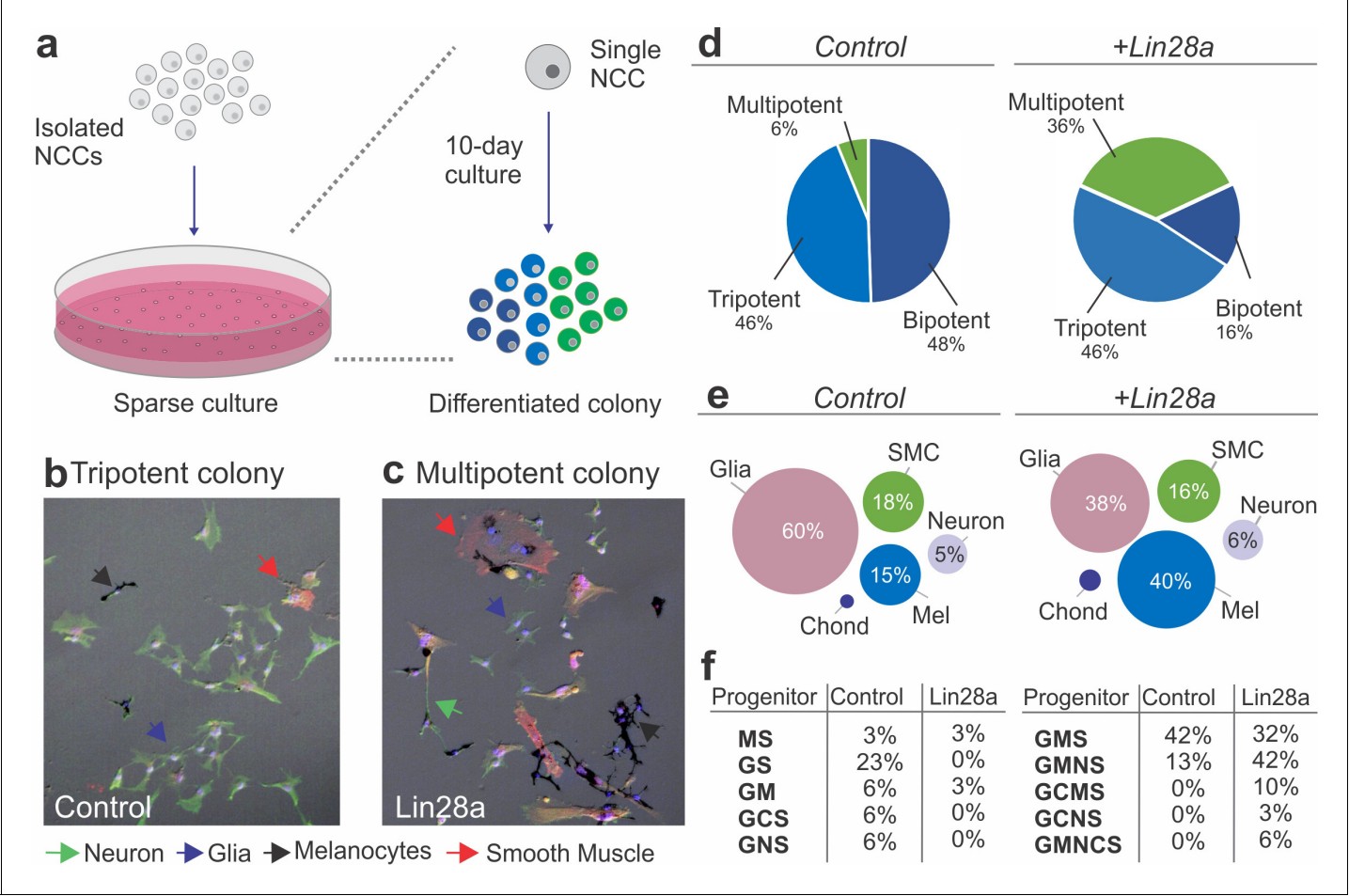

**Figure 2.** Lin28a modulates the developmental potential of neural crest cells. (a) Diagram showing the experimental design for neural crest single-cell clonal analysis. Quail neural crest cells transfected with control or *Lin28a* expression construct were isolated and plated sparsely to ensure the formation of clonal colonies. Cells were cultured for up to 10 days, after which the different cell types in each colony were identified by immunofluorescence. (b–d) Neural crest cells expressing higher levels of *Lin28a*, formed increased number of multipotent colonies. Representative images showing a tripotent colony derived from a neural crest cell transfected with control construct (b), and a multipotent colony derived from a single neural crest cell transfected with a *Lin28a* expression construct (+Lin28a) (c). (d) Pie-charts showing the percentage of bi-, tri-, and multipotent colonies formed by control and +Lin28a neural crest cells. (e) Representation of the frequency of different cell types observed in all colonies formed by control and +Lin28a neural crest cells. (f) Table listing the percentages of the different neural crest progenitor cells observed in control and +Lin28 a conditions. NCC: neural crest cell, N: neuron, G: glia, M: melanocyte, C: chondroblast, S: smooth muscle cell.

DOI: https://doi.org/10.7554/eLife.40556.008

The following source data is available for figure 2:

**Source data 1.** Quantitation of the different cell types in control vs +Lin28 neural crest derived colonies.
DOI: https://doi.org/10.7554/eLife.40556.009

*let-7*s through RT-PCR (*Figure 3f*, *Figure 3—source data 1*). Furthermore, transfection with a *let-7a* mimic molecule resulted in loss of FoxD3 +cells (*Figure 3g–h*) and reduced the expression of neural crest markers (*Figure 3i*), recapitulating the phenotype of *Lin28a* knockdown. This treatment did not alter proliferation or cell death, as evidenced by immunostaining with pH3 and Caspase-3 antibodies, respectively (*Figure 1—figure supplement 4d–e,h–i*).

To confirm that Lin28a regulates neural crest development predominantly via *let-7* inhibition (*Figure 3j*), we performed Lin28a knockdown and attempted to rescue the phenotype with a wild-type *Lin28a* construct, a *Lin28a* mutant unable to bind to *let-7* miRNAs (mCCHC *Lin28a*)(*Heo et al., 2008*) or a *let-7* sponge construct, which sequesters *let-7* molecules to decrease its activity (*Kumar et al., 2008*) (*Figure 3k–l*). The wild-type Lin28a protein was able to rescue the loss of neural

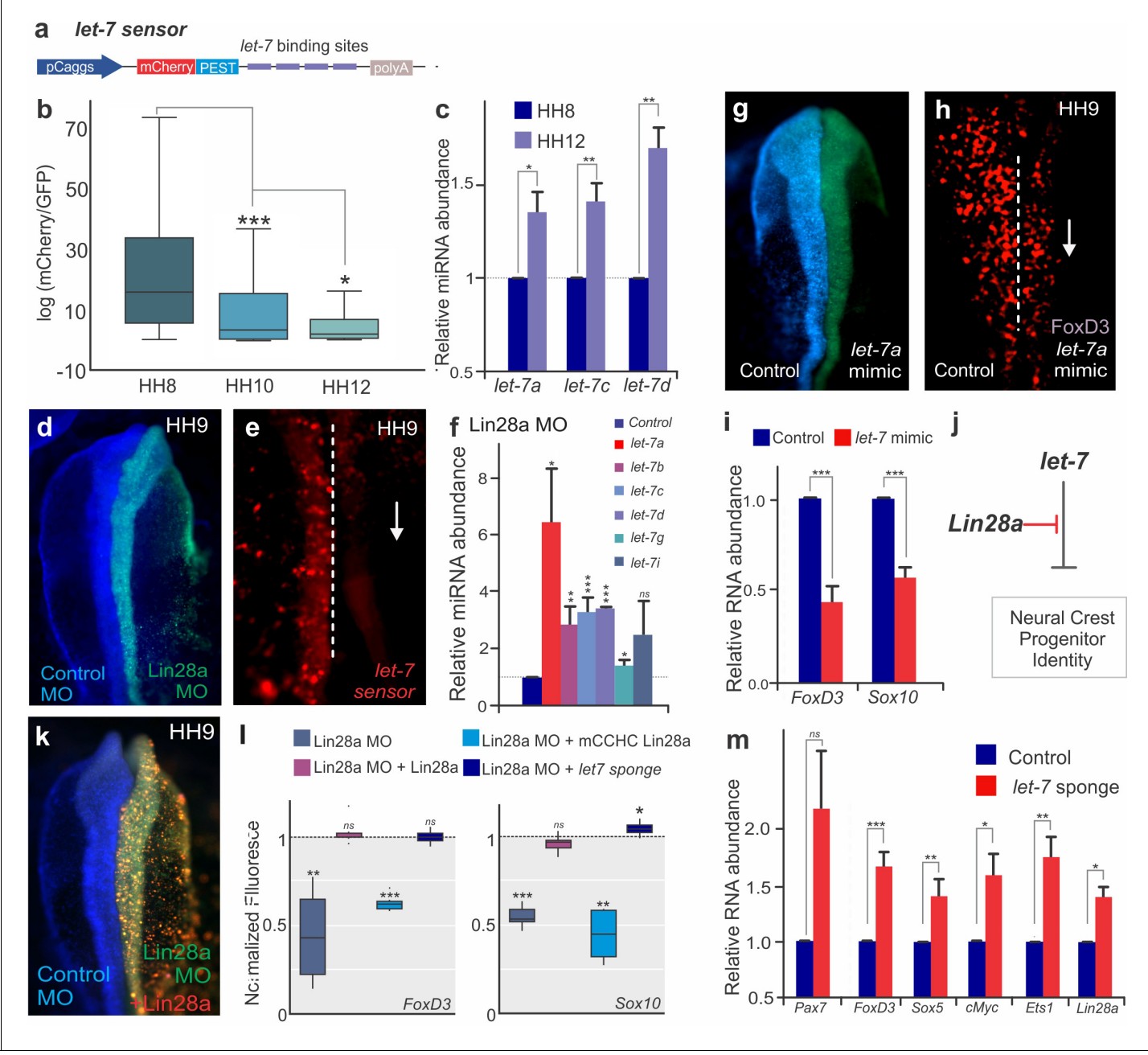

**Figure 3.** The Lin28/*let-7* axis modulates neural crest progenitor identity in vivo. (a) A schematic representation of the *let-7* sensor, which consists of several *let-7* binding sites downstream of destabilized mCherry fluorescent protein. (b–c) Activity of mature *let-7* miRNAs increase through neural crest development. (b) Boxplots showing mCherry/GFP fluorescence ratio, a readout of *let-7* sensor activity, in neural crest cells at different developmental stages. (c) RT-PCR for mature *let-7* family miRNAs comparing their levels in neural crest cells sorted from HH8 and HH12 embryos. (d–f) Loss of Lin28a results in increased activity of mature *let-7* miRNAs. (d) Whole mount view of an embryo bilaterally injected with control and *Lin28a* MO. (e) Representative image showing *let-7* sensor fluorescence in control vs *Lin28a* MO side of an embryo. Dotted line represents embryo midline. (f) RT-PCR for mature *let-7* family miRNAs, in the background of *Lin28a* knockdown. (g) Whole mount view of an embryo electroporated with control and let-7a mimic. (h) Immunohistochemistry for FoxD3 positive neural crest cells in the presence of *let-7a* mimic. Dotted line represents embryo midline. (i) Quantification of transcript levels of *FoxD3* and *Sox10*, in presence of increased *let-7a*. (j) Model for modulation of neural crest identity by the Lin28/*let-7* axis. (k) Representative dorsal view of an embryo electroporated with control MO (blue) on the left and *Lin28a* MO (green) co-injected with a *Lin28a* expression vector (red) on the right. (l) Boxplots showing the quantification of FoxD3 and Sox10 fluorescence in epistatic experiments, in which Lin28a Mo was co-electroporated with Lin28a expression vector, mCCHC Lin28a, and a *let-7* sponge construct. (m) Loss of *let-7* activity results in maintenance of multipotency genes in late neural crest cells. RT-PCR for *Pax7, FoxD3, Sox5, Myc, Ets1* and *Lin28a* comparing the expression of these genes in

*Figure 3 continued on next page*

*Figure 3 continued*

control vs late migratory neural crest cells expressing let-7 sponge construct. Error bars in (c), (f), (i) and (m) represent standard error. HH: Hamburger and Hamilton developmental stages, MO: Morpholino.

DOI: https://doi.org/10.7554/eLife.40556.010

The following source data is available for figure 3:

**Source data 1.** Data for the RT-PCR experiments shown in *Figure 3*, and quantitation of FoxD3 and Sox10 intensity in epistasis experiments.

DOI: https://doi.org/10.7554/eLife.40556.011

crest; however, the *mCCHC* Lin28a mutant protein could not restore FoxD3 or Sox10 expression. Furthermore, transfection with a *let-7* sponge construct, which reduces the levels of free mature *let-7*, recapitulated the wild-type Lin28a rescue (*Figure 3k–l*, *Figure 3—source data 1*). These results indicate that *Lin28a* regulates neural crest genes via the *let-7* dependent pathway, and suggest that *let-7* miRNAs mediate the silencing of neural crest genes observed during late migration (*Figure 1d*). To test if the increase in *let-7* levels (*Figure 3b–c*) results in silencing of early neural crest genes, we reduced the levels of mature *let-7s* using the sponge construct and quantified the expression of multipotency genes in late neural crest cells. RT-PCR analysis revealed that this inhibition of *let-7* activity results in maintenance of *Pax7*, *FoxD3*, *Sox5*, *cMyc*, and *Ets1*, recapitulating the effect of *Lin28a* overexpression (*Figure 1e*, *Figure 3m*). Consistent with published data showing that *Lin28a* is itself a *let-7* target (*Rybak et al., 2008*), and that these factors form a double-negative feedback loop, we also detected higher levels of the pluripotency factor in these cells (*Figure 3m*). Taken together, these experiments indicate that neural crest stem cell identity is regulated by the *Lin28a/let-7* axis, consistent with the possibility that the balance between these two factors underlies the collapse of the multipotency network observed during differentiation.

## Post-transcriptional silencing of the neural crest gene regulatory network

To test the global transcriptional effects of the disruption of Lin28a/*let-7* balance, we employed single embryo Nanostring analysis to assay the effects of *Lin28a* knockdown and *let-7* gain-of-function on the expression of ~100 genes involved in neural, placodal and neural crest development (*Simões-Costa et al., 2015*) (*Figure 4a–d*). Both Lin28a loss- and *let-7* gain-of-function recapitulated the changes that occur in late migratory neural crest cells (*Figure 1d*). By comparing control and targeted cells from individual embryos, we found that neural crest genes were strongly down-regulated in both treatments, suggesting that premature *let-7* activity has a systemic effect on the neural crest gene regulatory network (*Figure 4d*). Furthermore, genes that have been reported to modulate neural crest multipotency, such as *Sox10*, *FoxD3*, *Sox5*, *Ets1*, and *cMyc*, were particularly susceptible to changes in the Lin28a/*let-7* balance (*Figure 4d*, *Figure 4—source data 1*). We validated the Nanostring results by single-embryo RT-PCR (*Figure 4e*, *Figure 4—source data 1*), which additionally showed a strong downregulation of *Pax7* and *Tfap2b* genes (for which we lacked functional Nanostring probes) following manipulation of *let-7* levels. These results indicate the Lin28a/*let-7* axis regulates neural crest development by modulating the entire transcriptional network. Furthermore, the striking loss of stem cell genes (*Lignell et al., 2017*) observed in these experiments indicates that increased *let-7* activity suppresses multipotency and stemness in neural crest progenitors.

Next, we employed UTR-reporter assays to identify direct targets of *let-7* miRNAs in the early neural crest transcriptional network. 3' UTRs of six neural crest genes (*Figure 4—figure supplement 1a*) that were robustly affected in our functional assays (*Figure 4d–e*) were cloned downstream of a destabilized reporter gene (mCherry-PEST) and bilaterally transfected in chick embryos with or without a *let-7* mimic (*Figure 4f–h*). A similar construct driving GFP expression but lacking the 3' UTR was used as a transfection control. Flow cytometry analysis was used to compare reporter activity on each side of the same embryo (*Figure 4i*). The 3' UTR reporters for *FoxD3*, *Pax7*, and *Myc* (*Figure 4i–j*; *Figure 4—figure supplement 1c,f*), which have *let-7* target sites (*Figure 4—figure supplement 1a*), showed decreased activity when they were co-transfected with the *let-7* mimic. In contrast, the reporters for *Sox10*, *Sox8* and *Zic1* UTRs, which lack target sites for the microRNAs, were unaffected by the gain-of-function assay (*Figure 4j*, *Figure 4—figure supplement 1b,d–e*). To further test the importance of post-transcriptional regulation in the silencing of early neural crest genes,

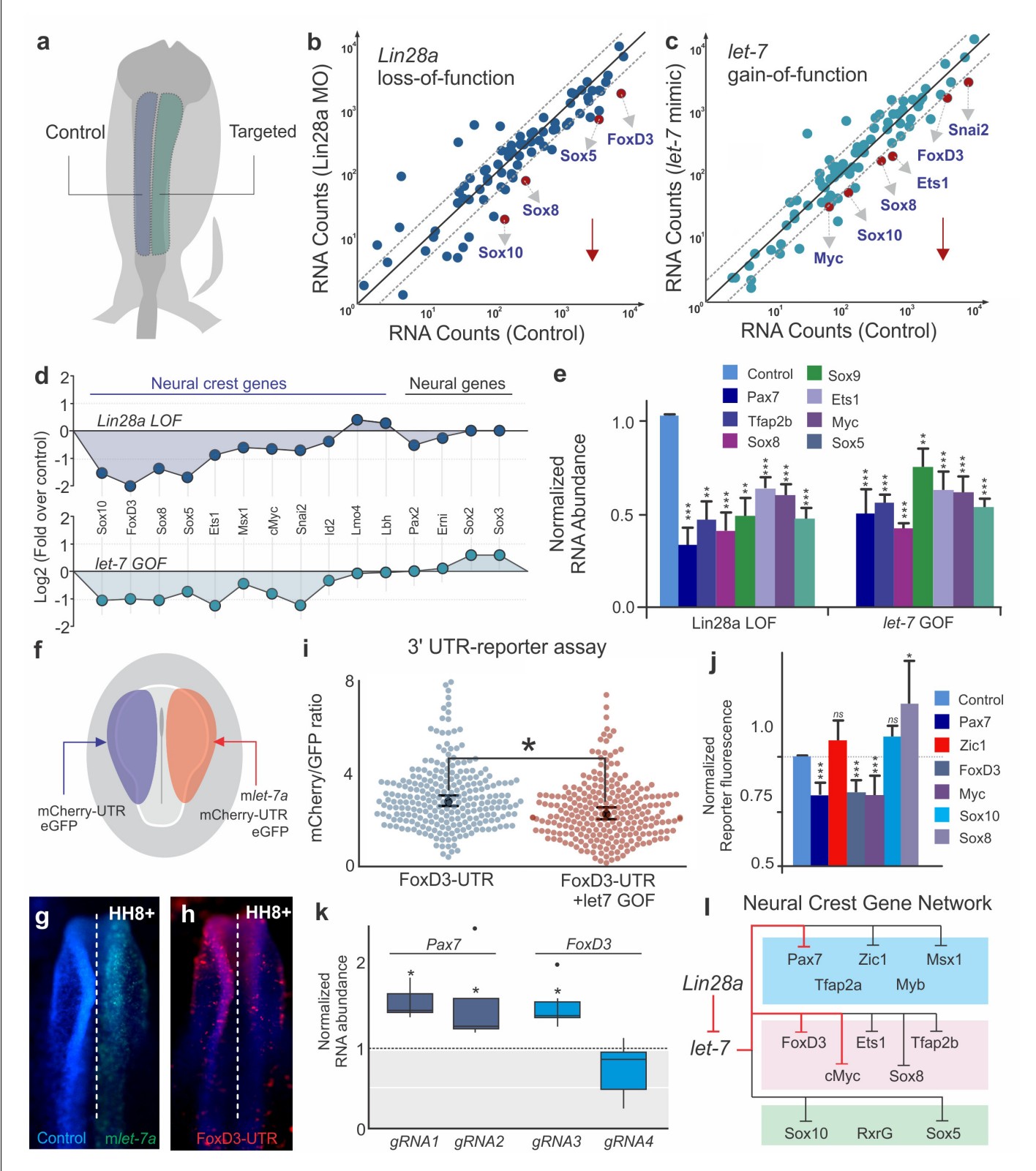

**Figure 4.** *let-7* directly targets multipotency circuits in the neural crest transcriptional network. (**a**) Control and targeted neural folds were dissected from the same embryo for Nanostring analysis. Comparison of transcript levels between (**b**) Control MO and *Lin28a* MO electroporated cells and (**c**) between control and *let-7* mimic injected cells. Genes below the diagonal dotted line were significantly downregulated (downward red arrow) in each condition. (**d**) Neural crest genes were similarly affected in *Lin28a* loss of function (LOF) and *let-7* gain-of function (GOF) assays. (**e**) RT-PCR performed
*Figure 4 continued on next page*

*Figure 4 continued*

with control and targeted neural folds to validate Nanostring results. (f) Electroporation scheme for in vivo 3'UTR reporter assay. Individual 3'UTR reporter constructs were co-injected with a control oligo (left) and a *let-7* mimic (right). Flow cytometry analysis was performed to measure mCherry and eGFP fluorescence of individual cells. (g–h) whole mount view of an embryo showing FoxD3-UTR reporter fluorescence in control vs *let-7* mimic transfected side of the embryo. (i) Representative scatter plots of FoxD3 UTR-reporter assay, showing the mCherry/GFP intensity ratio in cells analyzed from the control (gene-UTR) and *let-7a* mimic transfected (gene-UTR +*let7* GOF) sides of the same embryo. Each dot in the plot represents a single cell. (j) Average fold change in the ratio of mCherry/GFP intensity for each 3'UTR analyzed. (k) Quantification of fold change in *Pax7* and *Foxd3* transcript levels in late migratory neural crest cells when the *let-7* binding site on the 3'-UTR of these genes are targeted with specific gRNAs. *gRNA1* and *gRNA2* against Pax7 3'-UTR specifically targets the two let-7 binding sites, while *gRNA3* and *gRNA4* for FoxD3 3'-UTR targets a *let-7* binding site and another control region on the UTR, respectively. (l) Lin28/let-7 targets in the early neural crest transcriptional network, showing genes that are directly (red inhibitory lines) or indirectly (black inhibitory lines) affected by *let-7*. Error bars in (e) and (j) represent standard error and standard deviation respectively. MO: Morpholino, LOF: loss-of-function, GOF: gain-of-function.

DOI: https://doi.org/10.7554/eLife.40556.012

The following source data and figure supplement are available for figure 4:

**Source data 1.** Raw counts of the Nanostring experiment and the data for the RT-PCR experiments shown in *Figure 4*.
DOI: https://doi.org/10.7554/eLife.40556.014
**Figure supplement 1.** *let-7* miRNAs regulate 3'-UTRs of neural crest genes.
DOI: https://doi.org/10.7554/eLife.40556.013

we compared endogenous FoxD3 protein expression with the activity of the enhancer that controls its expression in migratory cells (*FoxD3NC2*) (*Simões-Costa et al., 2012*). We found that FoxD3 protein expression rapidly decreased during migration, while enhancer activity did not significantly change between neural crest cells close to the neural tube and those that had further migrated out (*Figure 4—figure supplement 1g–j*). This indicates that post-transcriptional regulation, which is absent from the enhancer construct, is necessary for timely silencing of components of the gene regulatory network.

To demonstrate the importance of *let-7* target sites for the endogenous regulation of neural crest genes, we employed CRISPR/Cas9 genome editing. A Cas9/eGFP expression vector containing gRNAs targeted to *let-7* sites within the 3'UTR of *Pax7* and *FoxD3* where transfected in gastrula stage chick embryos with bilateral electroporation (the control side was transfected with Cas9/eGFP only. Transfected cells from the control and experimental sides of single embryos were isolated with FACS at HH12, and analyzed with RT-PCR (*Figure 4—figure supplement 1k*). Targeting of individual *let-7* sites in both the *Pax7* and *FoxD3* loci resulted in a mild but consistent increase in the expression of these genes (*Figure 4k*), at the stages that they are normally downregulated; a control gRNA (gRNA4) targeting a region of the *FoxD3* UTR devoid of let-7 miRNAs sites had no effect on gene expression. These results indicate that *let-7* miRNAs orchestrate the silencing of progenitor cell identity by directly repressing critical network nodes (*Figure 4i*). We speculate that these inhibitory interactions propagate in a domino-like fashion throughout the network, resulting in its collapse (*Figure 4l*).

## Wnt signaling regulates multipotency in a position-dependent manner

The above results show that the dynamics of Lin28a/*let-7* activity regulate multiple targets of the early neural crest transcriptional network. To explore the upstream regulators of the Lin28a/*let-7* axis, we investigated the transcriptional regulation of *Lin28a* in neural crest cells. Assay for Transposase-Accessible Chromatin (ATAC-seq) performed in sorted neural crest cells revealed eleven non-coding regions of open chromatin in the *Lin28a* locus (*Figure 5—figure supplement 1a*). Transient transgenesis experiments in chick embryos showed that only one of these regions, located in the second intron of Lin28a (*Lin28E1*) (*Figure 5a*, *Figure 5—figure supplement 1b*) was able to drive reporter activity in the neural crest and ectoderm (*Figure 5b*). The sequence of this enhancer, which is conserved in amniotes, contains four TCF/LEF binding sites (*Figure 5—figure supplement 1c*), suggesting regulation by canonical Wnt signaling. Consistent with this, Lin28a protein expression was strongest in the dorsal neural tube, a known source of Wnt ligands (*Simões-Costa et al., 2015*; *Hollyday et al., 1995*) (*Figure 5c–d*). To test whether Wnt-signaling directly regulates *Lin28a* in neural crest cells, we first mutated the four TCF/LEF binding sites in *Lin28E1*. This resulted in complete loss of enhancer activity specifically in neural crest cells (*Figure 5e–f*), and also prevented enhancer

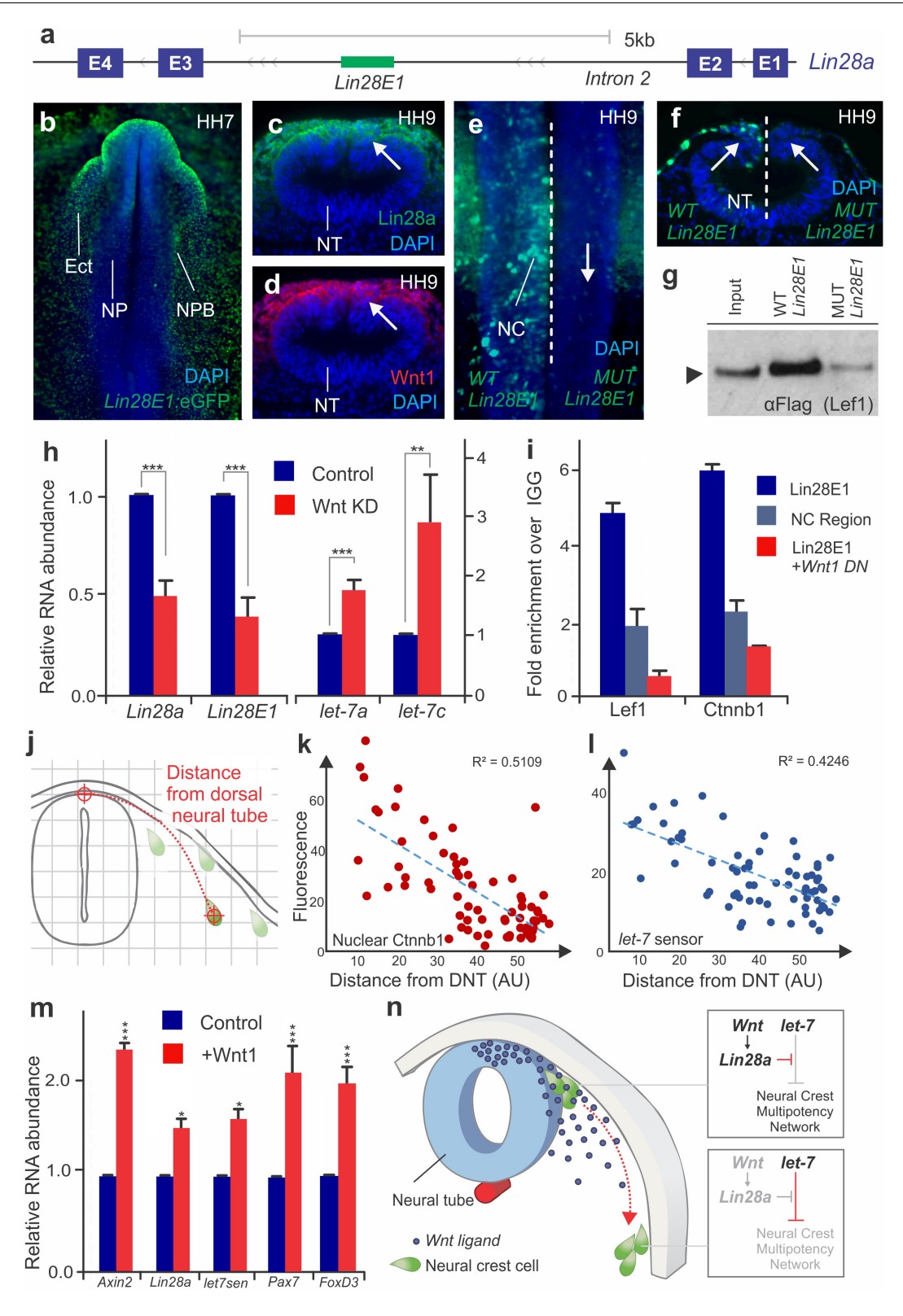

**Figure 5.** Positional information determines silencing of multipotency during neural crest migration (a) The avian *Lin28a* gene locus, showing the exonic (E1–E4) and intronic regions. *Lin28E1* is a 698 bp region within the second intron, which was identified as a *Lin28a* enhancer. (b) Expression pattern of a *Lin28E1* driven eGFP construct, which is active in the ectoderm (Ect) and at the neural plate border (NPB). Immunohistochemistry for Lin28a (c) and Wnt1 (d) on transverse sections of HH9 embryo (arrows point to the dorsal tube). (e–f) Comparison of *Lin28E1* (left) and *MUT Lin28E1* (right) reporter
*Figure 5 continued on next page*

*Figure 5 continued*

activity in a bilaterally electroporated embryo. Dorsal view of the head of a whole mount embryo (**e**) and transverse section (**f**, arrows point to neural crest cells). (**g**) Western blot for flag-Lef1 following enhancer pull down experiment with wild type and mutant *Lin28E1*. (**h**) RT-PCR for endogenous *Lin28a* mRNA, *Lin28E1* reporter and mature *let-7* miRNAs on combinatorial knockdown of Wnt1 and Wnt4 (**i**) Chromatin immunoprecipitation for Lef-1 and β-catenin (Ctnnb1), performed with neural folds of WT embryos, and embryos electroporated with Wnt dominant negative construct. (**j**) Diagram outlining the parameters for single cell measurements of *let-7* sensor fluorescence and nuclear β-catenin (Ctnnb1), as a function of the distance to the dorsal neural tube. Fluorescence intensity of nuclear β-catenin (**k**) and *let-7* sensor (**l**) is inversely correlated with the distance of the neural crest cell from the neural tube. In both graphs, each dot represents a single cell. (**m**) Quantitative comparison of transcript levels of *Axin2*, *Lin28a*, *let-7* sensor, *Pax7* and *FoxD3* in control vs. Wnt1 overexpressing neural crest cells (**n**) Model summarizing the results. AU: arbitrary units, DN: dominant negative, DNT: dorsal neural tube, IGG: Immunoglobulin G, *Lin28E1*: Lin28a Enhancer 1, Neg region: negative control region. Error bars in (**h**) and (**m**) represent standard error and error bars in (**i**) reflect standard deviation between technical replicates.

DOI: https://doi.org/10.7554/eLife.40556.015

The following source data and figure supplements are available for figure 5:

**Source data 1.** Data for the RT-PCR experiemtns shown in *Figure 5* and quantification of single cell measurements of *let-7*-sensor activity and nuclear b-catenin.

DOI: https://doi.org/10.7554/eLife.40556.018

**Figure supplement 1.** Identification of a *Lin28a* intronic enhancer.

DOI: https://doi.org/10.7554/eLife.40556.016

**Figure supplement 2.** Prolonged Wnt signaling affects neural crest differentiation (**a**) Normalized expression levels of differentiation genes *Alx1*, *Runx2*, and *Barx2* in migratory neural crest cells in different stages of development.

DOI: https://doi.org/10.7554/eLife.40556.017

association with Wnt-effector Lef1 in enhancer pull-down experiments (*Figure 5g*). Second, we conducted loss-of-function experiments by disrupting Wnt signaling using morpholinos targeting two Wnt ligands expressed in the dorsal neural folds, Wnt1 and Wnt4 (*Simões-Costa et al., 2015*). This knockdown reduced expression of both endogenous *Lin28a* and *Lin28E1* activity (*Figure 5h*). Moreover, loss of Wnt signaling resulted in increased levels of mature *let-7s* (*Figure 5h*). Finally, chromatin immunoprecipitation (ChIP) revealed that Lef1 and nuclear b-catenin (Ctnnb1) are associated with *Lin28E1* in neural crest cells, indicating that canonical Wnts are directly regulating *Lin28a*. This interaction is dependent on Wnt activity, as binding to the enhancer was lost in embryos transfected with a *Wnt1* dominant negative construct (*Figure 5i*).

Taken together, these results indicate that a Wnt-Lin28a/*let-7* regulatory circuit controls neural crest stem cell identity during differentiation. We hypothesize that a Wnt-mediated stem cell niche in the dorsal neural tube activates *Lin28a* expression in neural crest cells, thereby protecting the stem cell regulatory network from *let-7* mediated repression. If this assumption is correct, neural crest cells should exhibit a reduction in Wnt-activation and an increase in *let-7* activity as they migrate away from the neural tube. To test this, we measured nuclear β-catenin (*Ctnnb1)* and *let-7* sensor fluorescence as a function of distance from the neural tube in single migratory neural crest cells (*Figure 5j*). Consistent with our prediction, the analysis revealed a decrease in Wnt activity and increased *let-7* mediated repression during migration (*Figure 5k–l*, *Figure 5—source data 1*). Based on these results, we propose that the topological relationship between a neural crest cell and the Wnt niche determines the balance of Lin28a/*let-7* activity, which in turn modulates the early transcriptional network. Thus, we expanded the Wnt niche beyond the dorsal neural tube by ectopically expressing Wnt1 in migratory neural crest cells (*Figure 5m*, increased *Axin2* transcript levels confirmed over-activation of the pathway) (*Jho et al., 2002*). As predicted, we found that migrating neural crest cells constitutively expressing Wnt1 have higher levels of *Lin28a* and lower *let-7* activity. The expansion of the Wnt niche also prevents silencing of early neural crest factors *Pax7* and *FoxD3* (*Figure 5m*). Furthermore, this maintenance of stem identity resulted in suppression of differentiation, as Wnt gain-of-function resulted in lower expression levels of drivers of ectomenchymal differentiation *Runx2*, *Alx1*, and *Barx2* (*Figure 5—figure supplement 2a–b*). These findings show that manipulation of the Wnt-Lin28a/*let-7* regulatory circuit impacts both progenitor identity and the onset of differentiation, indicating that this mechanism controls the transition between these two states.

## Discussion

The neural crest is a migratory and multipotent cell type that undergoes extensive regulatory changes during differentiation. Hence, it is a powerful *in vivo* model to explore how environmental cues and transcriptional identity are integrated during cell state transitions. In this study, we examine neural crest development to characterize a mechanism linking positional information and the remodeling of gene regulatory networks that control multipotency. According to our model (*Figure 5n*), high levels of Wnt ligands produced by the dorsal neural tube activate *Lin28a* transcription in neural crest stem cells. The high levels of the *Lin28a* in turn inhibit *let-7* activity, protecting the neural crest transcriptional network from repression by these miRNAs. As neural crest cells migrate away from the Wnt source, *Lin28* levels are significantly reduced, resulting in an increase of mature *let-7* levels and subsequent repression of multipotency factors. This inhibition of crucial network nodes results in silencing of the neural crest gene regulatory network and loss of stem cell identity.

Wnt is a major modulator of neural crest identity, acting reiteratively during the formation and differentiation of this cell type (*Raible and Ragland, 2005*). Our results indicate that canonical Wnt signaling acts via the Lin28a/*let-7* axis to promote neural crest multipotency. Our model elucidates how this sginaling system provides developing cells with spatial information during cell fate restriction (*Loh et al., 2016*). As neural crest cells migrate away from the dorsal neural tube, we observe a gradual decrease in the activation of the pathway (*Figure 5k*), which underlies silencing of neural crest progenitor identity (*Figure 1d*). These spatial dynamics suggest an interesting parallel to the classic niche models in stem cell biology, in which the microenvironment provides signals that endow progenitor cells with broad potential (*Morrison and Spradling, 2008*). Our results indicate that the dorsal neural tube is an important signaling center in the embryo, acting to maintain neural crest stem cells in a multipotent state. The Lin28a/*let-7* axis is a crucial component of this mechanism. Little was known about the role of *Lin28a* in neural crest development before our analysis; previous work showed that *Lin28a* and its paralog, *Lin28b*, are dynamically expressed during amniote embryonic development (*Yokoyama et al., 2008*), and in vitro experiments suggest that these factors play a role in neurogliogenesis (*Balzer et al., 2010*). Here we identify an essential function of *Lin28a* factor during the early stages neural crest development, which is consistent with its role as a regulator of pluripotency in stem cells and cancer (*Shyh-Chang and Daley, 2013*).

We postulate that this regulatory mechanism is independent of the role of Wnts in neural crest induction and specification (*Simões-Costa et al., 2015*; *García-Castro et al., 2002*), and its later function as a driver of differentiation into melanocytes (*Dorsky et al., 1998*) and sensory neurons (*Lee et al., 2004*). Neural crest cells display specific responses to Wnt manipulations performed at distinct stages of development, both before (*Simões-Costa et al., 2015*; *García-Castro et al., 2002*) and after specification (*Hari et al., 2012*). Thus, we believe our model is compatible with the previous studies that show a requirement of canonical Wnts in cell fate decisions. While we still have a limited understanding of the mechanisms that compartmentalize the distinct functions of the pathway, time-controlled studies have shown that timing of intracellular response is crucial for specificity (*Hari et al., 2012*). Our results suggest that Lin28 activation by Wnts is established very early in neural crest progenitors and that this interaction is lost as the response to the pathway is attenuated during migration. In birds, neural crest cells form two waves of migration, with the melanocytic subpopulation delaminating later than the chondrocytic and neural progenitors (*Erickson and Goins, 1995*; *Le Douarin and Kalcheim, 1999*). While we hypothesize that the Wnt-Lin28/*let-7* circuit operates in all neural crest cells, further experiments will be necessary to clarify how the Lin28/*let-7* axis operates in these subpopulations, which can differentiate at distinct positions relative to the dorsal neural tube.

Our findings also clarify how intricate regulatory programs can be rapidly silenced during cells state transitions. Studies in the neural crest gene network have revealed numerous positive interactions that act to stabilize progenitor cell identity (*Simões-Costa and Bronner, 2015*). Indeed, the presence of positive regulatory loops is a common feature of developmental regulatory networks (*Davidson, 2010*; *Davidson and Levine, 2008*). Nevertheless, the identity of a progenitor cell is transient by definition, and thus the network has to collapse as cells differentiate. This rapid silencing of progenitor identity during differentiation is evident in our quantitative analysis (*Figure 1d*), which shows that the majority of the components of the network are only transiently expressed, being downregulated as the cells migrate away from the neural tube. Genes that are interlinked by positive

interactions are still progressively lost as cells migrate ventrally. For instance *Pax3/7* has been shown to activate itself to stabilize and maintain neural crest identity (*Plouhinec et al., 2014*). Similarly, *Tfap2a*, *Sox9* and *Snai2* form a positive feedback loop in neural crest cells, (*Luo et al., 2003*) which may explain the rapid increase in expression of these factors during neural crest specification. *SoxE* factors are also known to partake and auto-regulatory loops in multiple contexts (*Mead et al., 2013*; *O'Donnell et al., 2006*; *Honoré et al., 2003*). Thus, there is a disconnect between the logic encoded in network architecture (*Simões-Costa and Bronner, 2015*) and plasticity of cell identity observed in vivo.

We address this by identifying a post-transcriptional mechanism that is able to override positive regulatory interactions, silencing the neural crest gene regulatory network. Our experiments indicate that in the absence of *Lin28a*, the expression of mature *let-7* microRNAs increases dramatically (*Figure 3c–e*), reaching levels that are sufficient to inhibit neural crest stem cell identity. This is due to the fact that the *let-7* target genes identified in our UTR-reporter assay represent critical nodes of the network that are involved in the regulation of multiple neural crest genes (*Simões-Costa and Bronner, 2015*). *Pax3/7*, *FoxD3* and *cMyc* are stem cell factors and *bona fide* markers of neural crest cells, which play central roles in induction and specification (*Dottori et al., 2001*; *Krishnakumar et al., 2016*; *Basch et al., 2006*; *Kerosuo and Bronner, 2016*). These factors are also part of a group of genes that define neural crest stem cell identity (*Lignell et al., 2017*) located within the Wnt niche. Our demonstration that a Wnt-Lin28a/*let-7* regulatory circuit is able to modulate this regulatory program highlights how gene networks can be extensively remodeled during cell state transitions.

# Materials and methods

## Key resources table

| Reagent type (species) or resource | Designation | Source or reference | Identifiers | Additional information |
|---|---|---|---|---|
| Antibody | Mouse Anti-Lin28a | DSHB Cat # IE2 | RRID:AB_2618825 | IHC 1:4 |
| Antibody | Goat Anti-Sox10 | R and D Systems Cat# AF2864 | RRID:AB_442208 | IHC 1:50 |
| Antibody | Mouse Anti-Pax7 | DSHB | RRID:AB_528428 | IHC 1:4 |
| Antibody | Rabbit Anti-Cas9 | Takara | Cat #:632607 | IHC 1:200 |
| Antibody | Rabbit Anti-Wnt1 | Abcam Cat #: ab15251 | RRID:AB_301792 | IHC 1:200 |
| Antibody | Mouse Anti-Ctnnb1 | BD Biosciences Cat :610154 | RRID:AB_397555 | IHC 1:100 |
| Antibody | Mouse Anti-Tuj1 | BioLegend, Cat #:801202 | RRID:AB_10063408 | IHC 1:200 |
| Antibody | Rabbit Anti-pH3(S10) | Abcam Cat# ab47297 | RRID:AB_880448 | IHC 1:200 |
| Antibody | Rabbit Anti Caspase-3 | R and D Systems Cat #: AF835 | RRID:AB_2243952 | IHC 1:100 |
| Antibody | Rabbit Anti-mCherry | Abcam Cat # ab167453 | RRID:AB_2571870 | IHC 1:200 |
| Recombinant DNA Reagent | pRNA-U6-*let-7*-sponge | Addgene | plasmid # 35664 | |
| Recombinant DNA Reagent | pCAGGS-let-7-mCherry-PEST sensor | this paper | | Detailed in Materials and methods section |
| Recombinant DNA Reagent | pX333 vector | Addgene | plasmid # 64073 | |
| Recombinant DNA Reagent | pCAGGS-Lin28a-H2B-RFP | ths paper | | Detailed in Materials and methods section |
| Recombinant DNA Reagent | pCAGGS-Lin28a-mCCHC-H2B-RFP | this paper | | Detailed in Materials and methods section |

*Continued on next page*

*Continued*

| Reagent type (species) or resource | Designation | Source or reference | Identifiers | Additional information |
|---|---|---|---|---|
| Recombinant DNA Reagent | pTK-*Tfap2aE1*-GFP | this paper | | Detailed in Materials and methods section |
| Recombinant DNA Reagent | pTK-*Lin28E1*-GFP | this paper | | Detailed in Materials and methods section |
| Sequence based reagent | Lin28a morpholino | Genetools (this paper) | | 5'-AAACAGACCCCA TCCCGACACTCGC-3' |
| Sequence based reagent | Wnt1 morpholino | Genetools (this paper) | | 5'-GATGATGCCCCTA CGGAGCGGGAAT-3' |
| Sequence based reagent | Wnt4 morpholino | Genetools (this paper) | | 5'-GCGCAGGAAATAC TCCGGGCTCATC-3' |
| Sequence based reagent | *Lin28a* gRNA1 | this paper | | 5'-ACCGATACCCTCAA AGCTGGCCGAGG-3' |
| Sequence based reagent | *Lin28a* gRNA2 | this paper | | 5'-CACCGCGCTTGCAA ATTCCGAGTTGTGG-3' |
| Sequence based reagent | *Lln28a DsiRNA1* | IDT (this paper) | | 5'GCCGUUGAAUU CACCUUCAAGAAAT-3' |
| Sequence based reagent | *Lin28a DsiRNA2* | IDT (this paper) | | 5'-GGGGUCUGUUU CCAACCAGCAGUTT-3' |
| Sequence based reagent | *LIN28b DsiRNA1* | IDT (this paper) | | 5'-GUGGAAUUUAC UUACAAGAAAUCTT-3' |
| Sequence based reagent | *Lin28B DsiRNA2* | IDT (this paper) | | 5'-AAGCUUACAUGG AAGGAUUUAGAA-3' |
| Commercial assay or kit | qScript microRNA cDNA Synthesis Kit | Quanta Biosciences | Cat #: 95107–025 | |
| Commercial assay or kit | Rneasy Plus Micro kit | Qiagen | Cat #: 74034 | |
| Commercial assay or kit | Power SYBR Green Cells-to-CT Kit | Thermo Fisher | Cat #: A35379 | |
| Software, algorithm | ImageJ/Fiji | NIH | | |
| Software, algorithm | *nSolver* | Nanostring technologies | | |

## Embryo collection and fixation

Fertilized chicken eggs (Leghorn White) were purchased from University of Connecticut (Department of Animal Science). Eggs were incubated at 37°C until embryos reached the desired developmental stage. Embryos were collected and cultured according to the EC protocol (*Chapman et al., 2001*) and staged based on Hamburger and Hamilton (*Hamburger and Hamilton, 1951*). For immunohistochemistry, embryos were fixed with phosphate buffer (PB) containing 4% PFA for 20 min at room temperature (RT) and processed immediately. For in situ hybridization, embryos were fixed in phosphate buffer saline (PBS) containing 4% paraformaldehyde (PFA) for 2 hr at RT or overnight at 4°C. Following fixation, embryos were dissected, washed with PBST, dehydrated and stored in methanol at −20°C. Whole-mount in-situ hybridization was performed as previously described (*Wilkinson, 1992*). For double in-situ hybridization, we used the Tyramide TSA system from Perking Elmer (TSA Plus Cyanine 5 and Fluorescein, NEL754001KT) as previously described (*Denkers et al., 2004*).

## Embryo transfection and perturbation experiments

Chick embryos at HH4-5 were transfected with morpholinos, DsiRNAs and DNA constructs by *ex ovo electroporation*, as previously described (*Simões-Costa et al., 2015*). Briefly, morpholinos/DsiRNAs or DNA expression vectors were injected between the epiblast and vitelline membrane of dissected embryos and electroporated with platinum electrodes (five 50 ms pulses of 5.1V, with an

interval of 100 ms between pulses) (*Sauka-Spengler and Barembaum, 2008*). In all gene knockdown and overexpression experiments, the embryos were injected bilaterally with the control reagent on the left side and the targeted reagent on the right side. Whole embryo injections were performed for enhancer analysis and neural crest sorting experiments. Following electroporation, embryos were cultured in albumin at 37°C until they reached the desired developmental stages. Embryo survival was >90% and all embryos were screened to ensure that only uniformly electroporated, healthy embryos were used for further analysis. *Lin28a* knockdown was performed using FITC labeled translation-blocking morpholino (5'-AAACAGACCCCATCCCGACACTCGC-3') (GeneTools). Both control and *Lin28a* morpholinos were injected at a final concentration of 1.5 mM, supplemented with 1 μg/μl of carrier DNA and 10 mM Tris pH 8.0. For *Lin28a* and *Lin28b* loss of function experiments using DsiRNAs, the control and targeted DsiRNAs (IDT) were injected at a final concentration 20 μM. The sequence of the DsiRNAs used are as follows:

Lin28a DsiRNA1: 5'GCCGUUGAAUUCACCUUCAAGAAAT-3'
Lin28a DsiRNA2: 5'-GGGGUCUGUUUCCAACCAGCAGUTT-3'
Lin28b DsiRNA1: 5'-GUGGAAUUUACUUACAAGAAAUCTT-3'
Lin28b DsiRNA2: 5'-AAGCUUACAUGGAAGGAUUUAGAA-3'.

The gga-*let-7-a-5p* mimic (miScript miRNA mimic, Qiagen, MSY0001101) was electroporated at a concentration of 100 μM, with1μg/μl of carrier DNA and 10 mM Tris pH8.0, to facilitate entry into cells.

## Immunohistochemistry

For whole mount immunohistochemistry, embryos were dissected from the filter paper after fixation and washed in TBS containing 0.1% Triton and 1% DMSO (TBTD). Embryos were blocked for 2 hr in TBTD supplemented with 10% donkey serum and incubated in primary antibody diluted in blocking solution, overnight at 4°C. The following primary antibodies were used: anti-Lin28a mouse monoclonal (DSHB, 1:4), anti-Sox10, goat polyclonal (R and D Systems, AF2864, 1:50), anti-Foxd3, rabbit polyclonal (1:200, gift from Patricia Labosky), anti-Pax7, mouse (DSHB AB528428, 1:4), anti-Cas9, rabbit polyclonal (Takara 632607, 1:200) anti-WNT1, rabbit polyclonal (Abcam, ab15251, 1:200), anti-Ctnnb1 (β-catenin) mouse monoclonal (BD Transduction Laboratories, 610154 1:100), anti-Tuj1 (BioLegend, 801202,1:200), anti-pH3(S10) (Abcam, ab47297, 1:200), anti-Caspase3 (R and D Systems, AF835, 1:100) and anti-mCherry, rabbit polyclonal (Abcam, ab167453, 1:200). Secondary antibodies used included donkey anti-mouse/goat/rabbit IgG conjugated with Alexa Fluor 350/488/568/647 or goat anti-mouse Alexa 633 (Molecular Probes, 1:3000). Quantification of fluorescence for phenotype quantification in gain- and loss-of-function studies was performed with ImageJ.

## Cryosectioning

Fixed embryos were washed in 5% sucrose for 3 hr at RT, and in 15% sucrose solution overnight at 4°C. Next, they were incubated in 7.5% porcine gelatin for 3 hr at 37°C, embedded in silicone molds, snap frozen in liquid nitrogen and stored at −80C. 5–10 μM sections were obtained using the Cryo-Star NX50 (Thermo Fisher). For imaging, the slides were immersed in PBST at 42°C for 15 mins for gelatin removal, washed in PBS and mounted with Fluoromount-G (Southern Biotech, 0100–01).

## Expression vectors

The *Lin28a* expression construct was assembled by insertion of the full-length coding sequence of avian *Lin28a* in a pCI-H2B-RFP backbone. The coding sequence of *Lin28a* was PCR amplified from an HH8 cDNA library. To generate the mCCHC mutant version of the *Lin28a* expression construct, we introduced two mutations: H147A and H169A in the CCHC domain of the protein, which has been previously reported to abolish *Lin28a* binding to the stem-loop region of *pre-let-7* miRNAs (*Heo et al., 2008*). The pRNA-U6-*let-7* sponge construct was a gift from Philip Zamore (Addgene plasmid # 35664). The *let-7* sensor was constructed by cloning the *let-7* sponge sequence (amplified from the pRNA-U6-*let-7* sponge plasmid), downstream of a destabilized mCherry coding sequence (mCherry-PEST), in a pCAGGS backbone. The specificity of the sensor was assayed by electropotation of a let-7a mimic molecular, which resulted in a strong loss of mCherry expression. All expression vectors were sequenced to ensure that no additional mutations were present.

## Nanostring analysis

To identify the genes regulated by the Lin28a/*let-7* circuit, we performed Nanostring analysis in two experimental conditions: *Lin28a* morpholino-mediated knockdown and *let-7* mimic treatment. Stage HH4 chick embryos were electroporated with control morpholino on the left side, and with *Lin28a* morpholino or *let-7* mimic on the right side. Both morpholinos were injected at a concentration of 1.5 mM and the *let-7* mimic was diluted to a final concentration of 100 µM/µl. Post-electroporation, embryos were incubated at 37°C for ~12 hr, until they reached stage HH9. The control and targeted dorsal neural folds of embryos were microdissected and lysed in RNAqueous lysis buffer (RNAqueous-Micro Kit, AM1931). RNA lysates were hybridized at 65°C for 12 hr to a Nanostring probe set containing ~100 probes for neural crest, placodal and neural genes (*Simões-Costa et al., 2015*). Analysis of Nanostring data was performed with the nSolver software.

## 3'UTR reporter assay

3'UTR reporter constructs for *FoxD3*, *Pax7*, *Myc*, *Sox10*, *Zic1*, and *Sox8*, were built by amplifying the 3'UTR regions of these genes (as annotated in UCSC genome browser, Galgal 5.0) from an HH8 cDNA library prepared with oligo dT primers. Each 3'UTR was fused to a destabilized mCherry reporter (mCherry-PEST), in a pCAAGS vector backbone. In gastrulating embryos, 1 µg/µl of a 3'UTR reporter construct was transfected in the control side of the embryo (left) and co-transfected with *let-7a* mimic in the experimental side of the embryo (left). As a transfection control,1 µg/µl of an eGFP expression construct built with the same vector backbone but lacking the 3'-UTR regions, was co-injected with the mCherry reporters. After incubation of embryos at 37°C for 12 hr, control and *let-7* mimic transfected halves of the same embryo were dissected and processed independently for flow cytometry. Fluorescent intensity of mCherry and GFP in dual positive cells were quantified, and the ratio of mCherry/GFP intensity was used as a parameter for measuring reporter activity.

## Embryo dissociation and cell sorting

For isolation of neural crest cells, embryos were transfected with 1 µg/µl of an enhancer of the *Tfap2a* gene (*Attanasio et al., 2013*) (*Tfap2aE1*) cloned into PTK-eGFP (*Uchikawa et al., 2003*). To obtain neural crest cells from different stages, embryos were cultured until HH6 (8 hr), HH8 (~11 hr), HH10 (13–14 hr), HH12 (~18 hr) and HH14 (~23 hr), and screened for robust GFP expression in neural crest cells. Embryo heads were dissected in Ringers solution, washed with dPBS and incubated in Accumax (Accutase SCR006) cell dissociation solution, for 40 min at RT under mild agitation. Following this, dissociated cells were passed through a cell strainer (Pluriselect USA, Mini Cell Strainer II, 45-09840-50) and centrifuged at 400 g for 10 min. The supernatant was carefully discarded and cells were resuspended in 200 µl of HANKS buffer supplemented with 0.5% BSA. At least 1500 GFP +and GFP- cells from each stage were sorted directly into 50 µl of lysis buffer from Power SYBR Green Cells-to-CT Kit (ThermoFisher, 4402953) using BD AriaFusion cell sorter. To assay for the effects of *Lin28a* overexpression or *let-7* sponge over-expression in neural crest cells, embryos were bilaterally electroporated with *Tfap2aE1* on the left and with *Tfap2aE1* + Lin28a-RFP/*Tfap2aE1* + *let-7* sponge expression vector on the right. Following incubation at 37°C for 18–19 hr, each half of the head of individual embryos were dissected and processed separately for FACS sorting. GFP+/ RFP +populations of cells were sorted from the experimental side of the embryo, while GFP +neural crest cells were collected from the control side. We employed RT-PCR to compare gene expression levels between control and targeted cells obtained from the same embryo.

## Quantitative reverse transcription PCR (RT-PCR)

To quantify changes in gene expression caused by perturbation/reprogramming experiments, we microdissected single neural folds from control and targeted side of the embryo, which were subsequently lysed in lysis buffer from Power SYBR Green Cells-to-CT Kit. RNA extraction and cDNA preparation were performed according to the kit's protocol). RT-PCR was performed using Power Sybr Green PCR master mix (Thermo Fisher, 4368577) in an ABI viia7 RT-PCR machine. Ct values of all genes were normalized to reference gene *hprt* and expressed as a fold change compared to the control sample.

## Cornish pasty culture of chick embryos

To assess the long term consequences of *Lin28a* knockdown, cornish pasty culture (*Nagai et al., 2011*) was performed with HH4 chick embryos bilaterally transfected with Lin28 MO. Following electroporation, embryos were transferred to Panett Compton media (12 ml of Solution1 +18 ml of Solution 2 + 270 ml of dH2O), released from the filter paper and folded along the anterior-posterior axis with the dorsal side out. Embryos were allowed to rest in Panett-Compton solution for about 30 mins. The excess extra-embryonic membrane was cut with fine surgical scissors, and the embryos were transferred to a media composed of 2:1 ratio of Albumin to Panett Compton solution. Embryos were incubated in this media at 37°C for 48–50 hr until they reached HH15.

## Quantification of mature let-7

To measure levels of mature *let-7* miRNAs, RNA was extracted from dissected control and Lin28aMO-targeted neural folds with RNeasy Plus Micro kit (Qiagen, 74034), following the guidelines for small RNA extraction. Poly(A) tailing and cDNA synthesis were performed using the qScript$^{TM}$ microRNA cDNA Synthesis Kit (Quanta Biosciences, 95107–025). RT-PCR for individual *let-7*s was done as suggested by the kit, with mature miRNA-specific primers, and a universal primer against the poly-A tail. Ct values were normalized to 18S rRNA and expressed as a fold change compared to the control sample.

## CRISPR-Cas9 mediated knockdown of gene expression

To knock down *Lin28a* using CRISPR-Cas9, gRNAs targeting the first exon of *Lin28a* were designed using online resources (crispr.mit.edu). A combination of two gRNAs was cloned downstream of the U6 promoters in the pX333 vector (a gift from Dr. Andrea Ventura, AddGene plasmid #64073) (*Maddalo et al., 2014*). To assay for the effects of *Lin28a* knockdown, HH4 embryos were bilaterally electroporated with empty *Cas9* vector on the left and with Cas9 +Lin28 a gRNA construct on the right. The embryos were incubated at 37°C until they reached HH9 +when they were stained with FoxD3 (*Mundell and Labosky, 2011*) and Sox10 (R and D Systems, AF2864) antibodies, sectioned and analyzed for phenotype. Individual FoxD3 and Sox10 +cells were counted from multiple sections obtained from three bilaterally transfected embryos. For disrupting *let-7* binding sites on 3'-UTR of *FoxD3* and *Pax7*, a unique gRNA was designed for each of the sites, as well as for a control region in the 3'-UTR of FoxD3 which did not contain *let-7* binding sites. The gRNAs were individually cloned under an U6-promoter in a modified pX333 vector, which had a GFP sequence cloned downstream of the Cas9 (Cas9-GFP). To assay for the effect of disruption of *let-7* binding sites on FoxD3 and Pax7, HH4 embryos were electroporated with empty Cas9-GFP vector on the left and with Cas9-GFP + UTR gRNA construct on the right. The embryos were allowed to develop until HH12, after which the control and targeted halves of the head of individual embryos were dissected separately, and FACS sorted for GFP +cells. Finally, we employed RT-PCR to measure the expression of *FoxD3* or *Pax7*, in cells obtained from the control vs. experimental side.

## Single-cell clonal analysis

Single-cell clonal analysis to assay for neural crest multipotency, for performed as described previously (*Lahav et al., 1998*). Briefly, HH4 quail embryos were injected with a control (pCI:H2B-RFP) or a *Lin28a* O/E (pCI:LIN28A-H2B-RFP) construct. The embryos were incubated at 37°C to develop until the 6-somite stage (HH9-) and screened for robust RFP expression. To isolate neural crest cells, we dissected 6–8 neural folds from control and Lin28a o/e quail embryos, and plated them on collagen coated tissue-culture dish containing 10%FBS-DMEM media. The explants were incubated for ~36 hr (at 37°C and 5% $CO_2$ conditions), until neural crest cells had migrated out and a halo of cells was visible around the explanted neural folds. The remaining neural fold tissue was removed, and the migratory neural crest cells were dissociated with Accumax. The cells were resuspended in fresh media (10% FBS-DMEM +2% Chicken Embryo Extract), and sparsely plated on collagen coated 6-well plates. After 3–4 hr, once the cells had attached and spread out, the plates were screened to make sure that the >90% of the plated cells were isolated, with only 1 cell/field visible using a 10X objective. The cells were allowed to differentiate over a period of 10 days, after which the different cell types were assayed using immunofluorescence. Immunofluorescence of neural crest clones was performed as described previously. Briefly, the cells were fixed in 4% PFA at RT for 10 mins.

Following fixation, the cells were permeabilized using 0.1% NP-40 solution in PBS at 37C for 30 mins. Next, the cells were blocked in 1%BSA solution at 37C for 30 mins, after which they were incubated with primary antibody cocktail at 1 hr at 37 C. The primary antibodies used for detecting different cell types was as follows: anti-SMA for myofibroblast (rabbit, Abcam), anti-Runx2 for cartilage (mouse IgG2a, DSHB), anti-GFAP for glia (rabbit, Abcam), anti-Neurofilament for neurons (mouse IgG2a, Biolegend) and anti-MelEM for melanocytes (mouse IgG1a, DSHB). Following incubation with primary antibody, the cells where washed and incubated with corresponding secondary antibodies for 90 mins at 37°C. Finally, the cells were washed, stained with DAPI and imaged using a Nikon eclipse inverted microscope. A total of 50 colonies per condition were scored for developmental potential (bipotent, tripotent or multipotent); a subset of these (30 per condition) were analyzed for cell composition to identify the progenitor time.

## let-7 sensor activity assay

For single-cell Quantification of sensor activity, the *let-7* sensor (mCherry) construct was co-transfected with *Tfap2aE1*(GFP) in HH4 embryos. Embryos were incubated until desired stages, and dissected heads were processed (as described above) for flow cytometry. Fluorescent intensity of mCherry was measured in GFP positive neural crest cells, and mCherry/GFP intensity ratio in each cell was used as a readout of sensor activity.

## Wnt loss- and gain-of-function

Wnt signaling was disrupted using two different strategies: combined inhibition of Wnt1 and Wnt4 with morpholinos (Wnt1: 5'-GATGATGCCCCTACGGAGCGGGAAT-3', Wnt 4: 5'-GCGCAGGAAA TACTCCGGGCTCATC-3') (*Simões-Costa et al., 2015*), and using a Wnt dominant negative vector (*García-Castro et al., 2002*). Morpholinos targeted to Wnt ligands were used at a concentration of 1.1 µM each, and the Wnt dominant negative construct was electroporated at 1 µg/µl. For activation of Wnt signaling, we employed a vector driving expression of Wnt1, which was electroporated at a concentration of 1 µg/µl.

## Enhancer analysis

To identify the enhancer controlling *Lin28a* expression, we cloned 2–3 kb DNA fragments from the *Lin28a* locus that contained regions of open chromatin as determined by ATAC-seq. We tested eleven ATAC peaks in the proximity of the *Lin28a* gene locus (Galgal 4.0 chr23:169040–169093; chr23:169493–169873; chr23:173387–173487; chr23:173838–174004; chr23:175014–175120; chr23:176664–176839; chr23:176849–177557; chr23:177927–178209; chr23:179032–179308; chr23:183167–183754; chr23:184717–184992). Peaks in close proximity were grouped and as a result, we cloned six regions in ptk-eGFP vector backbone (*Figure 5—figure supplement 1a*). These putative enhancer constructs were electroporated (at a concentration of 1 µg/ µl) in HH4 embryos and analyzed for eGFP expression at later stages. The region 39.1 (chr23:176849–177557) which is evolutionarily conserved in amniotes and was subsequently identified as the *Lin28a* enhancer, was henceforth referred to as *Lin28E1*. Computational analysis of the *Lin28E1* sequence with Jaspar (*Mathelier et al., 2016*) (jaspar.genereg.net) revealed the presence of four TCF/LEF binding sites. The mutant *Lin28E1* construct was constructed by replacing the 4 TCF/LEF1 binding sites in the *Lin28E1* construct with a repetitive 'CTCTCT' sequence of the same length as the binding sites. To assay for *Lin28E1* activity, the transcript levels of the enhancer-driven GFP were measured by RT-PCR, and GFP expression was normalized to *hprt* levels.

## Chromatin immunoprecipitation

For each experiment, chromatin was isolated from 16 cranial neural folds dissected from HH8-9 embryos in Ringer's solution. Immunoprecipitation was performed as described (*Simões-Costa et al., 2014*) using the *Lef1* (Millipore, #17–604) and *Ctnnb1* (BD Biosciences, #610154) antibodies and normal mouse IgGs (Millipore, #17–604) as controls. In *Wnt* dominant negative assays, HH5 embryos were electroporated with a *Wnt* dominant negative construct (*García-Castro et al., 2002*); at stages HH8-9, cranial neural folds were dissected. Chromatin isolation and immunoprecipitation were performed as previously described (*García-Castro et al., 2002*).

## Enhancer pull down

For enhancer pull-down experiments, HH4 embryos were electroporated with a Flag-tagged *Lef1* construct cloned in a pCAAGS-H2B-RFP backbone. Embryos were incubated until stage HH9. After electroporation efficiency was confirmed, embryos were dissected in Ringer's solution (n = 4 embryos per sample). Nuclear protein extracts were obtained as previously described (*Simões-Costa et al., 2015*). *Lin28E1*, and *MUT Lin28E1* were PCR amplified from pTK-eGFP vectors using a biotinylated forward primer (5'-AAAATAGGCTGTCCCCAGTG-3') and an untagged reverse primer (5'-ATATTTCTTCCGGGGACACC-3'). Immobilization of nucleic acids was performed using Dyna-beads MyOne Streptavidin T1 (Invitrogen, #2023-11-30) following manufacturer's protocol. For enhancer pull down, nuclear protein extracts were diluted in 10 mM Tris-HCl, 1 mM EDTA, 0.5 mM EGTA, 10% Glycerol, 0.25% NP-40 supplemented with 10 ug Poly(dI-dC) and incubated with biotiny-lated DNA coated Dynabeads for 90 min in rotation at 4°C. Magnetics beads were washed four times in with Washing Buffer (10 mM Tris-HCl, 1 mM EDTA, 0.5 mM EGTA, 100 mM NaCl, 10% Glyc-erol, 0.25% NP-40), and proteins eluted in RIPA modified buffer containing 1x Sample Reducing Agent and 1x LDS Sample Buffer (Invitrogen, #B0009 and #B0007) for 15 min at 80°C and 1400 rpm. Proteins were separated by electrophoresis on Bolt 4–12% Bis-Tris Plus mini gels (Invitrogen, #NW04120BOX), followed by immunoblotting on nitrocellulose membranes using Monoclonal anti FLAG antibodies (Sigma, #A8592).

## Quantification of wnt and let-7 activity in single neural crest cells

To measure levels of activation of the canonical Wnt pathway and *let-7* sensor activity in single neural crest cells, we conducted confocal microscopy of 5 μm embryo midbrain sections immunostained for β-catenin (Ctnnb1) and mCherry (*let-7* sensor). For image analysis, ImageJ software was used to quantify fluorescence intensity of individual neural crest cells within sections. To estimate *let-7* activity in neural crest cells, we measured mCherry intensity in *Tfap2aE1* positive cells. The perimeter of each neural crest cell was defined by expression of the enhancer. To quantify the activity of the canonical Wnt pathway, we measured the presence of β-catenin in nuclei of neural crest cells (DAPI was used to define the nucleus). The distance between each cell examined, and a fixed point on the dorsal neural tube was also measured. The distance from the dorsal neural tube was plotted linearly vs the fluorescence intensity of nuclear β-catenin or vs the *let-7* sensor intensity. A line of best fit was applied to the data to examine the correlation between migration and the levels of canonical Wnt and *let-7* activity.

## Quantification of NC2-mCherry-PEST reporter activity and FoxD3 protein levels in migrating neural crest cells

For assessing the correlation between enhancer activity and protein levels of FoxD3, HH4 chick embryos were electroporated with a *FoxD3- NC2* enhancer driven mCherry-PEST reporter construct, which is active in cranial migratory NC cells (*Simões-Costa et al., 2012*). These embryos were subse-quently incubated until HH12, and whole embryo immunostaining was performed for FoxD3. The embryos were embedded and cryosectioned, and single cell measurement of enhancer reporter fluorescence and the FoxD3 antibody was performed with 5 μm embryo midbrain sections. Image analysis and quantification was performed using ImageJ as described above.

## Statistical analysis

At least 10 embryos were analyzed in immunohistochemistry and in-situ hybridization assays per-formed downstream of genetic perturbations experiments. For 3'-UTR reporter assay and *let-7* sen-sor activity assay, five embryos were analyzed per reporter construct or per developmental stage, respectively. ChIP experiments were repeated three times, and the results of a representative exper-iment are shown in the figures. To assess the long term consequence of Lin28a knockdown, Cornish pasty culture and subsequent analysis of trigeminal ganglia formation was performed for atleast six embryos. Single-cell quantification of canonical Wnt and *let-7* activity was performed in ~100 cells in three embryos, with consistent results. The Nanostring experiments were performed with 3 repli-cates of the *let-7* gain-of-function (3 experimental and 3 controls) and 2 replicates of *Lin28a* loss-of-function experimental conditions (2 experimental and 2 controls). The *n* values and *p* values of all quantitative experiments are listed in *Supplementary file 1*. Student's t-test (one-tailed) was

performed to calculate p-values and p<0.05 were considered to be significant. Mann-Whitney test was used to calculate p-values for 3'-UTR reporter and *let-7* sensor activity assay, given the non-parametric and non-Gaussian distribution of intensity values of cells analyzed in the assays.

## Acknowledgments

We are indebted to Carol Bayles and Adam Paul Wojno for cell-sorting assistance at the BRC Flow Cytometry Cell Sorting Facility at Cornel University. We thank Dr. Rebecca M Williams for support in imaging acquisition through the BRC Imaging Facility, and Carolina Purcell for assistance with the long-term culture system for avian embryos. This work was supported by NIH grant R00DE024232, the Meinig Family Foundation, and a Basil O'Connor Starter Scholar Award to MS-C.

## Additional information

### Funding

| Funder | Grant reference number | Author |
|---|---|---|
| National Institute of Dental and Craniofacial Research | R00DE024232 | Marcos Simoes-Costa |
| March of Dimes Foundation | | Marcos Simoes-Costa |

The funders had no role in study design, data collection and interpretation, or the decision to submit the work for publication.

### Author contributions

Debadrita Bhattacharya, Conceptualization, Data curation, Investigation, Methodology; Megan Rothstein, Data curation, Investigation, Methodology; Ana Paula Azambuja, Data curation, Formal analysis, Investigation, Methodology; Marcos Simoes-Costa, Conceptualization, Data curation, Formal analysis, Supervision, Funding acquisition, Validation, Investigation, Visualization, Methodology, Writing—original draft, Project administration, Writing—review and editing

### Author ORCIDs

Marcos Simoes-Costa (iD) http://orcid.org/0000-0003-1452-7068

### Decision letter and Author response

Decision letter https://doi.org/10.7554/eLife.40556.022
Author response https://doi.org/10.7554/eLife.40556.023

## Additional files

### Supplementary files

• Supplementary file 1. Table listing the number of biological replicates analyzed the calculated p-values for the quantitative experiments performed in this paper. The corresponding figure number for each experiment is also included in the table.
DOI: https://doi.org/10.7554/eLife.40556.019

• Transparent reporting form
DOI: https://doi.org/10.7554/eLife.40556.020

### Data availability

All data generated for this study are included in the manuscript and supporting files. Source data files have been provided for all figures.

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
