## [Decision Letter]

Thank you for submitting your article "Control of neural crest multipotency by Wnt signaling and the Lin28/*let-7* axis" for consideration by *eLife*. Your article has been reviewed by three peer reviewers, and the evaluation has been overseen by a Reviewing Editor and Didier Stainier as the Senior Editor. The following individuals involved in review of your submission have agreed to reveal their identity: Charles K Kaufman (Reviewer #1); Robert A Cornell (Reviewer #2).

Overall, this is a very well done study that sheds important new light onto the factors that regulate neural crest multipotency. It will be useful for others both in and out of this field in terms of how these transcriptional programs can be suppressed to allow differentiation to occur properly.

We did have some relatively minor concerns that we hope you can address. The reviewers have discussed the reviews with one another and the Reviewing Editor has drafted this decision to help you prepare a revised submission.

Major comments:

1) *Lin28a* vs. *Lin28b*

Vertebrates usually have two Lin28 proteins, *Lin28a* and *Lin28b*. The authors do not indicate which ortholog is being investigated in this study, although it appears to be *Lin28a*. If true, the data presented on knockdown of *Lin28a* in chick is inconsistent with genetic null mutants in mice, which are viable but have variable penetrance phenotypes in neural progenitor proliferation and cell cycle exit to produce smaller brains and overall body size. These data suggest that the *Lin28a* may be only important for chick neural crest development, as *Lin28a* does not appear to be required for neural crest development in mammals. The differences between mouse and chick *Lin28a* phenotypes could be due to species differences (sub-specialization in chick versus redundancy in mouse) or technical differences (transient knockdown versus genetic null alleles) or both. To address this, we suggest either or both of:

a) Analyzing *Lin28b* expression in chick embryos, with the expectation that it is not expressed in the neural tube/folds;b) Demonstrate the extent of knockdown or CRISPR mutagenesis for *Lin28a*, which would help mitigate the possibility that off-target effects that are inherent with using these reagents are contributing to the *Lin28a* chick phenotypes, thereby causing the discrepancy between species.

2) Neural crest TFs vs. differentiation markers after *Lin28* knockdown

The authors use a subset of neural crest transcription factors for qRT-PCR analysis in Figure 1 that are all downregulated during neural crest development and sensitive to enforced *Lin28* expression. These data should also include neural crest differentiation markers (e.g., for chondrocytes, pigment and neurons) to support the model that *Lin28* is inhibiting differentiation. If the authors have RNA-seq data available already, this could also assess in an unbiased manner the global impact of *Lin28* on neural crest and neural tube progenitor differentiation.

3) Given that *Lin28* is also expressed in the neural tube, and previous data demonstrates it is required for neural progenitor proliferation and cell cycle exit, an alternative explanation for the loss of neural crest markers *Foxd3* and *Sox10* after morpholino knockdown is loss of neural tube progenitors through inhibiting proliferation, cell cycle arrest or increased cell death. Along the same lines, the forced expression of the *let-7a* mimic could impact the proliferation or survival of neural tube/neural crest progenitors that leads to decreased *FoxD3* levels. The authors should assess proliferation and survival of neural tube and neural crest progenitors after these manipulations to determine if these features cause the loss of neural crest numbers at later stages.

4) Are there any physiological consequences in neural crest derivatives after manipulating the Lin28/*let-7* axis? The authors show data supporting an effect of *Lin28* or *let-7* on early neural crest markers, but does this actually translate to an impact of neural crest lineages at later time points, such as loss of cartilage elements and cranial ganglia?

---

## [Author Response]

Major comments:1) Lin28a vs. Lin28bVertebrates usually have two Lin28 proteins, Lin28a and Lin28b. The authors do not indicate which ortholog is being investigated in this study, although it appears to be Lin28a. If true, the data presented on knockdown of Lin28a in chick is inconsistent with genetic null mutants in mice, which are viable but have variable penetrance phenotypes in neural progenitor proliferation and cell cycle exit to produce smaller brains and overall body size. These data suggest that the Lin28a may be only important for chick neural crest development, as Lin28a does not appear to be required for neural crest development in mammals. The differences between mouse and chick Lin28a phenotypes could be due to species differences (sub-specialization in chick versus redundancy in mouse) or technical differences (transient knockdown versus genetic null alleles) or both. To address this, we suggest either or both of:a) Analyzing Lin28b expression in chick embryos, with the expectation that it is not expressed in the neural tube/folds;

We have adjusted the gene nomenclature used in the manuscript to clarify that we are investigating the expression and function of the *Lin28a* paralog. We agree with the reviewers that our results indicate that there is no functional redundancy between *Lin28a* and *Lin28b* in avian neural crest development. To investigate this, we quantified the expression of *Lin28b* in FACS-sorted avian neural crest cells from different stages of development. We found that *Lin28b* expression levels in neural crest cells are very low when compared to *Lin28a*, and that the former is not enriched in this cell population. Furthermore, unlike *Lin28a, Lin28b* expression levels do not change during the transition from multipotency to differentiation, remaining low throughout neural crest development. We have now included these results in Figure 1—figure supplement 1Q-R. We also included the results of *Lin28b* loss of function analysis (see point below), which shows that it is not required for neural crest specification.

b) Demonstrate the extent of knockdown or CRISPR mutagenesis for Lin28a, which would help mitigate the possibility that off-target effects that are inherent with using these reagents are contributing to the Lin28a chick phenotypes, thereby causing the discrepancy between species.

To minimize concerns about off-target effects, we include controls that show (i) strong down-regulation of the *Lin28a* protein after morpholino knockdown (Figure 1—figure supplement 3C-D) and (ii) robust rescue of the specification defect when a *Lin28a* expression vector is co-transfected with the morpholino (Figure 3L), which is the gold standard of specification control for morpholino loss of function experiments. Furthermore we now employ RNAi interference (DsiRNAs, IDT – Kim et al., 2005) as an additional strategy to knockdown *Lin28a* (Figure 1—figure supplement 3N). In these assays, we employed two distinct DsiRNAs to confirm that knockdown of *Lin28a* mRNA results in reduction of expression of neural crest markers. We also employed DsiRNA knockdown to demonstrate that *Lin28b* loss-of function does not hinder neural crest specification (Figure 1—figure supplement 3O), which is consistent with the scenario discussed above.

2) Neural crest TFs vs. differentiation markers after Lin28 knockdownThe authors use a subset of neural crest transcription factors for qRT-PCR analysis in Figure 1 that are all downregulated during neural crest development and sensitive to enforced Lin28 expression. These data should also include neural crest differentiation markers (e.g., for chondrocytes, pigment and neurons) to support the model that Lin28 is inhibiting differentiation. If the authors have RNA-seq data available already, this could also assess in an unbiased manner the global impact of Lin28 on neural crest and neural tube progenitor differentiation.

We thank the reviewers for this suggestion, which lead to new experiments that provided further support for our model. We isolated late migrating neural crest cells that have been transfected with a *Lin28a* expression constructs, and quantified the expression of multiple differentiation markers. The results show that sustained expression of *Lin28a* inhibits the expression of several differentiation markers of chondrocytic (*Runx2, Alx1* and *Barx2*), neuronal (*Tuj1, HuC,* and *Mash1)* and glial differentiation (*Fabp7* and *Gfap*). These results, which are now included in Figure 1—figure supplement 2I, are consistent with the hypothesis that *Lin28a* promotes multipotency at the expense of differentiation.

3) Given that Lin28 is also expressed in the neural tube, and previous data demonstrates it is required for neural progenitor proliferation and cell cycle exit, an alternative explanation for the loss of neural crest markers Foxd3 and Sox10 after morpholino knockdown is loss of neural tube progenitors through inhibiting proliferation, cell cycle arrest or increased cell death. Along the same lines, the forced expression of the let-7a mimic could impact the proliferation or survival of neural tube/neural crest progenitors that leads to decreased FoxD3 levels. The authors should assess proliferation and survival of neural tube and neural crest progenitors after these manipulations to determine if these features cause the loss of neural crest numbers at later stages.

To address this point, we assessed cell proliferation (with an Anti-phospho-Histone H3 antibody) and cell death (with a Caspase-3 antibody) in the neural tube, following treatment with the *Lin28a* morpholino and the *let-7a* mimic. Our results show that there is no significant change in proliferation or cell death in the neural tube or neural crest cells following these manipulations (Figure 1—figure supplement 4A-I). This indicates that the neural crest specification phenotypes we observe following these treatments are cell-autonomous. We believe discrepancy between these results and the changes in neural progenitor proliferation in the mouse *Lin28a* knockout model may be due to our stage-controlled loss-of-function strategies vs. a genetic null allele.

4) Are there any physiological consequences in neural crest derivatives after manipulating the Lin28/let-7 axis? The authors show data supporting an effect of Lin28 or let-7 on early neural crest markers, but does this actually translate to an impact of neural crest lineages at later time points, such as loss of cartilage elements and cranial ganglia?

We thank the reviewers for this suggestion. To address the consequences of *Lin28a* knockdown in neural crest derivatives we employed the Cornish pasty culture system (Nagai et al., 2011). This allowed us to culture morphant embryos until later stages of development (HH15), when we could assess how disruption of *Lin28a* expression affects the formation of the cranial ganglia. We found that the targeted side of the embryo had disorganized, smaller trigeminal ganglia – which in many instances lacked the maxillary and mandibular branches. This is consistent with previous findings that highlight the requirement of neural crest cells for timely ganglion condensation and accurate establishment of neuronal connections (Gammill et al., 2007; Stark et al., 1997). We have included these results in Figure 1—figure supplement 4J-O.

References:

L. S. Gammill, C. Gonzalez, M. Bronner-Fraser, Neuropilin 2/semaphorin 3F signaling is essential for cranial neural crest migration and trigeminal ganglion condensation. Developmental neurobiology 67, 47-56 (2007).

D. H. Kim et al., Synthetic dsRNA Dicer substrates enhance RNAi potency and efficacy. Nature biotechnology 23, 222-226 (2005).

H. Nagai, M. C. Lin, G. Sheng, A modified Cornish pasty method for ex ovo culture of the chick embryo. Genesis 49, 46-52 (2011).

M. R. Stark, J. Sechrist, M. Bronner-Fraser, C. Marcelle, Neural tube-ectoderm interactions are required for trigeminal placode formation. Development 124, 4287-4295 (1997).